

# The 3D $Q_P$ Model of the China Seismic Experiment Site (CSES-$Q$1.0) and Its Tectonic Implications

Mengqiao Duan[1] Lianqing Zhou[1*] Ying Fu[2] Yanru An[3] Jingqiong Yang[4] Xiaodong Zhang[1]

[1]Institute of Earthquake Forecasting, China Earthquake Administration, Beijing 100036, China
[2]Sichuan Earthquake Administration, Chengdu, 61004, China
[3]China Earthquake Networks Center, China Earthquake Administration, Beijing 100045, China
[4]Yunnan Earthquake Administration, Kunming, 650224, China

*Correspondence to*: Lianqing Zhou (zhoulq@ief.ac.cn)

**Abstract.** The Chuan-Dian region is located in the southeastern part of the geologically complex and seismically active
Tibetan Plateau. Since 2008, the Chuan-Dian region has experienced several major earthquakes, including the Wenchuan $M_S$
8.0, Lushan $M_S$ 7.0, and Jiuzhaigou $M_S$7.0, making it one of the areas with the most severe earthquake disasters. The China
Seismic Experimental Site (CSES) under construction in this area will deepen the understanding of the
preparation and generation of earthquakes and the disaster mechanisms, which can further enhance the defense capability
against earthquake risks. To build a world-class seismic experimental field, it is necessary to establish high-precision
medium structure models. Currently, several institutions have established high-resolution three-dimensional (3D) velocity
models in the CSES, but there is still a lack of high-resolution 3D attenuation ($\propto 1/Q$) structure models. Using the local
seismic tomography method, we obtain the highest resolution 3D $Q_P$ model in the CSES to date. Combining the existing
velocity models in the CSES with other geophysical and geochemical observations by predecessors, this study shows that the
$Q_P$ value anomalies along large fault zones and some basin areas are low, reflecting the high degree of medium
fragmentation in these areas, with thick sedimentary layers or rich in fluids. The high attenuation anomaly of the upper crust
dipping westward in the Tengchong volcanic characterizes the possible upward flow of deep-seated magma from west to
east. This study also reveals that most of earthquakes above magnitude 6 occurred in low attenuation zones or the boundary
areas of high-low attenuation anomalies. The source areas of the 2008 Wenchuan $M_S$ 8.0 earthquake and the 2013 Lushan
$M_S$ 7.0 earthquake were separated by a low attenuation area, and there is still a risk of major earthquakes in the future. The
3D attenuation model constructed in this study will provide a high-resolution reference model for seismological and
earthquake disaster research in the CSES.

## 1 Introduction

The China Seismic Experimental Site (CSES) started construction in 2018 (regional range: 97.5°-105.5°E, 21°-32°N), with a
total area of approximately 780,000 square kilometers. The regional scope includes the Chuan-Dian block (CDB) located on
the southeastern Tibet Plateau and its surrounding areas, with a complex tectonic environment and various fault systems such



as compression, shear, and tension. Multiple large thrust and strike-slip active fault zones have developed in the region, such as the Xianshuihe, Zemuhe, Xiaojiang, Red River, Longmenshan, Huayingshan, and Lijiang-Xiaojinhe fault zones. They divide the Chuan-Dian area into multiple active blocks, such as the Chuan-dian block, the Western Yunnan block, the Southern Yunnan block. (Zhang et al., 2003) (Fig. 1a). The collision and continuous convergence of the Indian and Eurasian plates led to strong crustal deformation and rapid surface uplift. This region is also one of the regions with the most frequent seismicity in Chinese Mainland, including both interplate and intraplate earthquakes. In the past 50 years, there have been an average of 14 earthquakes with a magnitude of 6.0 or above and 3 earthquakes with a magnitude of 7.0 or above every 10 years (Fig. 1b). Among them, the Wenchuan 8.0 earthquake on May 12, 2008, the Lushan 7.0 earthquake on April 20, 2013, the Ludian 6.5 earthquake on August 3, 2014, the Yangbi 6.4 earthquake on May 21, 2021, and the Luding 6.8 earthquake on September 5, 2022 all caused serious casualties and property losses. The seismogenic environment and mechanism in this region have always been a hot topic of discussion among scholars. Previously, various geophysical observations, such as low wave velocity (Yao et al., 2008; Yang et al., 2012, 2020; Bao et al., 2015; Zhang et al., 2020; Liu et al., 2021), high conductivity (Bai et al., 2010; Li et al., 2019), high heat flow (Hu et al., 2000; Jiang et al., 2019), strong attenuation (Zhao et al., 2013) and strong radial anisotropy (Bao et al., 2020) reveal the morphology and genesis of channel flows in the middle and lower crust. Among them, research on 3D velocity tomography is dominant, including 3D velocity structures based on body waves (Wu et al., 2013; Deng et al., 2021; Wang et al., 2015;Huang et al., 2018).Three dimensional $V_S$ structure of the crust to upper mantle based on surface waves and background noise (Wang and Gao, 2014; Yao et al., 2008; Shen et al., 2016; Fu et al., 2017; Qiao et al., 2018; Zheng et al., 2019; Yang et al., 2020).

To investigate the high-resolution subsurface medium structure of the CSES and understand the mechanisms of strong earthquakes in the region, seismologists have constructed various velocity models for CSES using various data, including body waves, surface waves, and ambient noise surface waves. For instance, Xin et al. (2019) and Han et al. (2022) successively used the inversion method of body waves and joint inversion method of body- and surface-wave to establish high-resolution lithospheric velocity structures for the Chinese mainland (USTClitho1.0, USTClitho2.0), with horizontal resolutions in the CSES ranging from 0.5° to 1°. Liu et al. (2021) and Liu et al. (2023) utilized joint body- and surface-wave travel-time inversion method to establish two community velocity models for CSES (SWChinaCVM-1.0, SWChinaCVM-2.0), with the maximum horizontal resolution of 0.2-0.3°, providing high-resolution community velocity models for geophysics research in the CSES area. The most recent study, based on the joint inversion of receiver functions and surface waves, constructed a 3D $V_P$ and $V_S$ model for CSES (CSES-VM1.0) with a maximum horizontal resolution of 0.25° (Wu et al., 2024). However, the velocity structure mainly reflects the elastic structure of the medium and lacks constraints on important inelastic properties during earthquake nucleation.



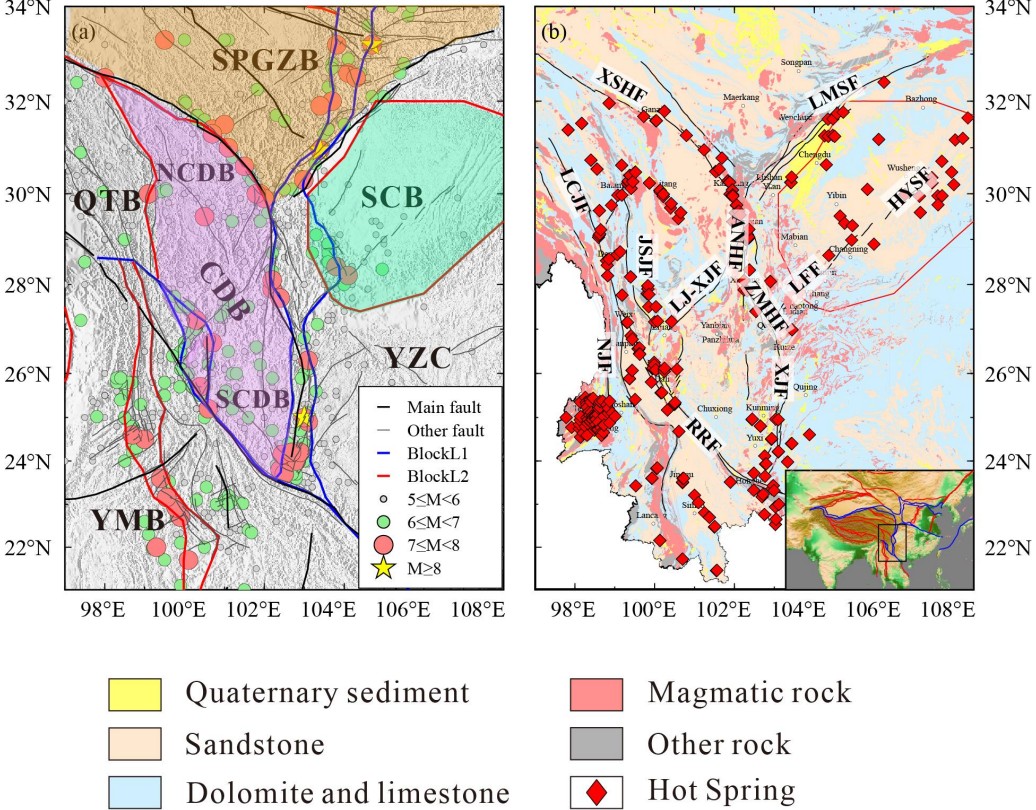

**Figure 1: The distribution map of tectonic structures, historical earthquakes, and lithology in the study area.**

**(a) Map of the tectonic structures and spatial distribution of earthquakes occurred from 780 BC to July 2023 in the CSES. SPGZB, Songpan-Ganze Block; QTB, Qiangtang Block; CDB, Chuan-Dian Block; NCDB, Southern Chuan-Dian Block; SCDB, Southern Chuan-Dian Block; YMB, Yunnan-Myanmar Block; SCB, Sichuan Basin; YZC, Yangtze Craton.**

**(b) Map of the spatial distribution of lithology and hot springs in the CSES. XSHF, Xianshuihe fault; LMSF, Longmenshan fault; ANHF, Anninghe fault; ZMHF, Zemuhe fault; XJF, Xiaojiang fault; HYSF, Huayingshan fault; LFF, Lianfeng fault; JSJF, Jinshajiang fault, LJ-XJHF, Lijiang-Xiaojinhe fault; LCJF, Lancangjiang fault; NJF, Nujiang fault.**

Seismic wave attenuation is an important parameter reflecting the inelastic properties of the medium. Seismic wave attenuation is typically extracted from the amplitudes of seismic waves (Pei et al., 2010) and is inversely proportional to the quality factor Q. The state of fractures, fluid migration, and thermal material upwelling in the medium can all cause variations in $Q$ (Yang et al., 2007; Zhu et al., 2013; Wang et al., 2017; Chen et al., 2021). Compared to seismic wave velocities, seismic wave attenuation is more sensitive to changes in fluids and temperature (Lin, 2014; Guo and Thurber, 2022). It can provide better constraints on the physical state of the medium and more important information for the thermal structure and dynamics of the lithosphere (Deng et al., 2021), and help us infer the permeability range of fluids and the the heterogeneity of thermal structures. Studies have also found that most moderate and strong earthquakes (magnitude 6 and above) in the Chuan-Dian region occur at the boundaries of high attenuation or high-low attenuation anomalies (Zhou et al.,



2008; Zhou et al., 2020). Therefore, attenuation structures can also provide scientific reference for the determination of the locations of large earthquakes.

At present, most attenuation models in the Chuan-Dian region are two-dimensional models of Lg waves or surface waves, which cannot accurately reveal the characteristics of attenuation in depth (Zhou et al., 2008; Zhao et al., 2013; Wei et al., 2019; Zhou et al., 2020). There are few published studies on 3D attenuation structures in the CSES and surrounding areas. Dai et al. (2020) only obtained a 3D body wave attenuation model for the southeastern part of the Chuan-Dian block, with low resolution. Tang et al. (2023) inverted a 3D shear wave attenuation model below a depth of 30 km, lacking the 3D

attenuation structure of the upper and middle crust. Liu et al. (2024) used teleseismic direct-P waves to obtain crustal and upper mantle  attenuation structure beneath the southeastern Tibetan Plateau, but there are only two layers at a depth of 100 km and 200 km. Therefore, CSES still lacks a high-precision crustal 3D attenuation model.

This paper collects a large number of seismic waveforms in CSES over the past decade and uses local earthquake tomography to construct a high-resolution 3D P-wave attenuation model for CSES. Combined with the existing 3D velocity

models in CSES, we can better understand the medium properties and seismogenic environment in CSES. It will provide a scientific basis for the crustal medium properties of the Tibetan Plateau, the mechanism of crustal material migration, and the assessment of the risk of large earthquakes.

## 2 Data and Method

### 2.1 Data

This study collates seismic catalogs, phase reports, and seismic waveforms from earthquakes with magnitudes greater than 1.5, recorded by a total of 582 stations from the Sichuan and Yunnan seismic networks within the study area (21°N -34°N, 97°E -108°E) since 2013. We select events that include at least six phases and picked phases with travel time residuals within ±2 seconds based on the travel time curves, and a total of 79,619 events from Sichuan and 39,668 events from Yunnan are selected. Due to the fact that earthquakes in Sichuan are more frequent and mainly concentrated around the

Longmenshan fault zone, the difference in ray density may result in significant variations in ray weights across different grids. To ensure the resolution of the $Q$ model and to uniformly cover the entire study area with rays, we clustered earthquakes in the Sichuan region, retaining only one earthquake within a 0.5 km range centered on each earthquake. Finally, we organized waveforms for 17,290 earthquakes in Sichuan and 18,488 earthquakes in Yunnan, with a total of 288,695 P-wave rays. The distribution of earthquakes before and after clustering in the region is shown in Fig. 2.



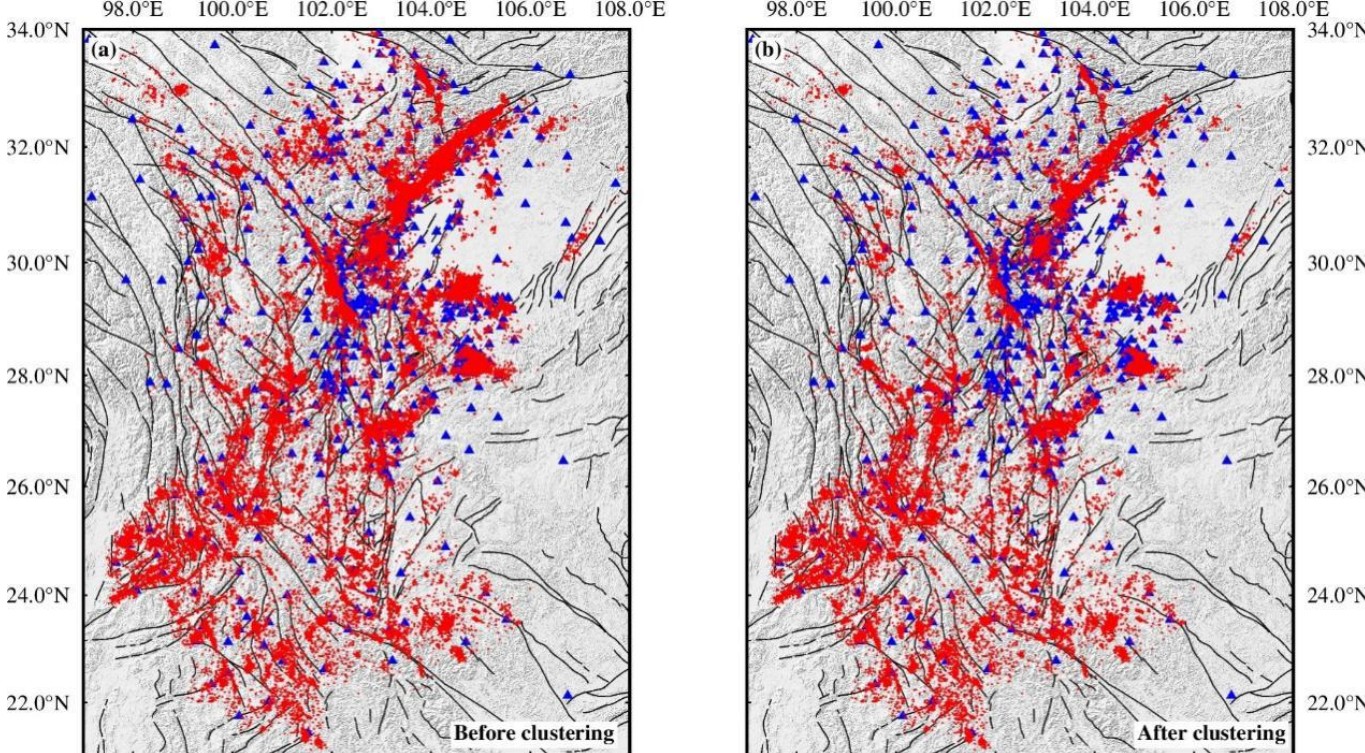

**Figure 2: Spatial distribution of earthquakes in the study area before (a) and after (b) clustering.**

**The red dots represent earthquakes, and the blue triangles represent stations.**

## 2.2 Method

The seismic wave attenuation conforms to the $\omega^2$ model (Brune, 1970), whose velocity amplitude spectrum can be expressed as follows:

$$A(f) = 2\pi f \cdot \frac{\Omega_0 \cdot f_c^2}{f_c^2 + f^2} e^{-\pi f t^*} \tag{1}$$

in which $\Omega_0$ is the spectral level at low frequency, $t^*$ is the whole path attenuation term $t/Q$, $f_c$ is the corner frequency of the event. Many laboratory and empirical studies found $Q = Q_0 f^\alpha$ (Karato and Spetzler, 1990; McNamara, 2000; Stachnik et al., 2004), where frequency-dependent factor $\alpha$ is usually in the range from 0 to 1. Some studies have found that the frequency-independent $Q$ images are quite similar to the frequency-dependent $Q$ images (Liu and Zhao, 2015; Wang et al., 2017). In this paper, we cannot solve for $\alpha$, referring to the method of Eberhart-Phillips and Chadwick (2002), and we assume $Q$ value is frequency-independent ($\alpha = 0$). The following steps are used to extract the $t^*$ within the frequency range of 2-20 Hz: (1) A flexible window method is used to cut the signal and noise windows of P waves (Zhou et al., 2011). For the P wave, the vertical component recording is selected. When the S-P arrival time difference is less than or equal to 2.56 s, the S-P arrival time difference is used as the signal window of the P wave, and the records whose S-P arrival time difference between is less





than 0.5 s are removed. When the S-P arrival time difference is greater than 2.56 s, 2.56 s is selected as the signal window of the P wave, and the noise window is set to 2.56 s before the P wave arrival time. (2) After removing the instrument response, the velocity spectra of the signal and noise are calculated, and the records with a signal-to-noise ratio greater than 2 are selected. (3) The corner frequency of each event is estimated using the grid search method. (4) An iterative algorithm is used to fit $\Omega_0$ and $t^*$. The $t^*$ values are weighted at a total of 5 levels (0, 1, 2, 3, 4) according to the fitting quality, in which 0, 1, 2, 3 represent the fit values with the fitting error less than 0.1 s, 0.2 s, 0.3 s and 0.4 s, respectively. Ultimately, we retain data with $t^*$ levels less than 4, and ensure that each event has at least 3 $t^*$ data. we obtained a total of 176,105 $t^*$ data for P waves. Below is an example of $t^*$ fitting. (Fig. 3).

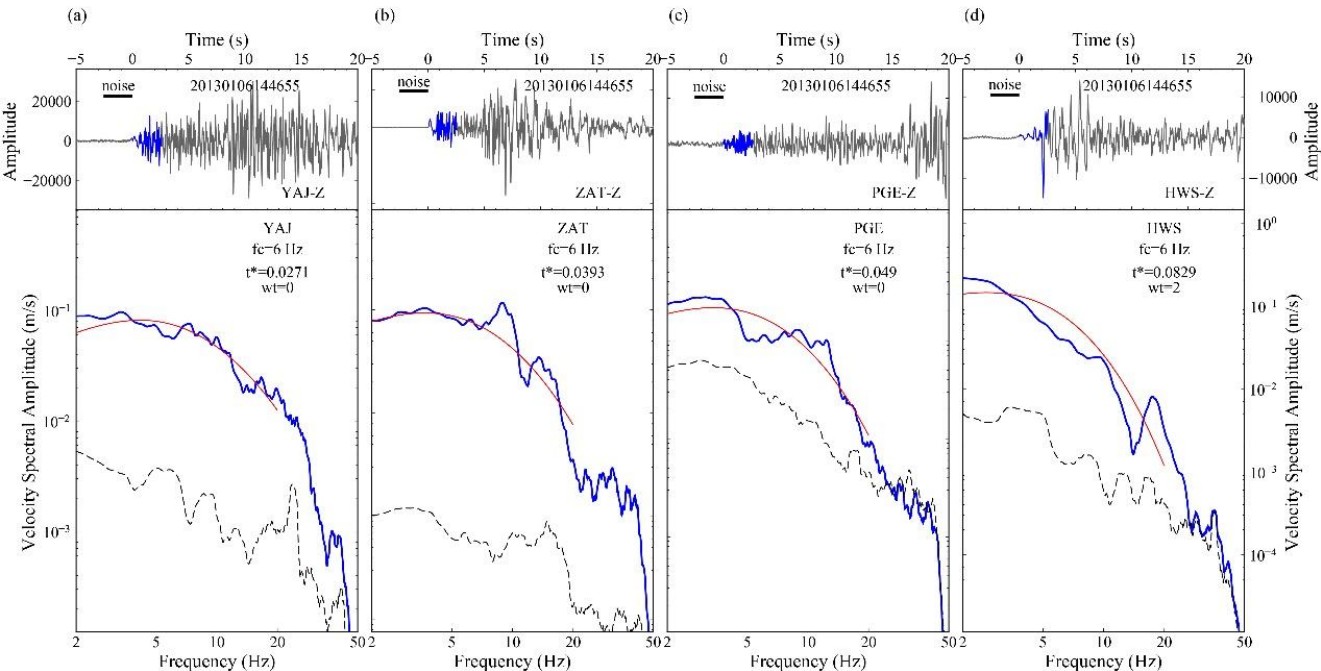





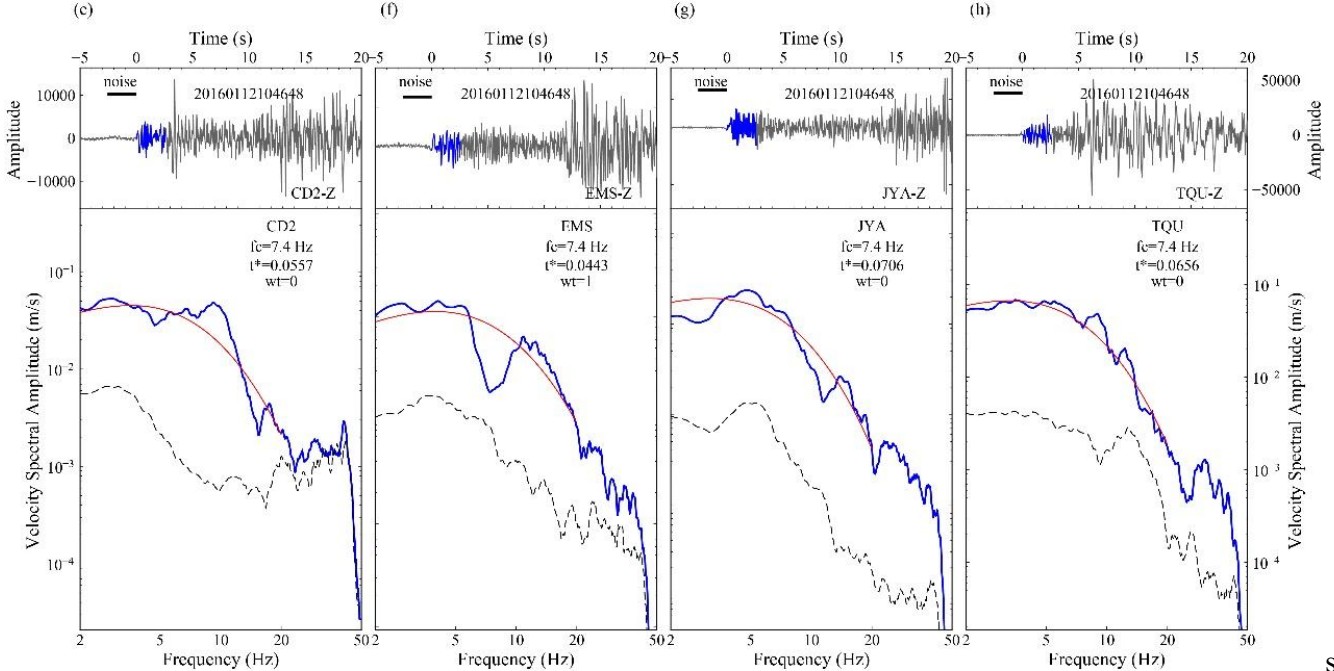

130                                                                                                                                    s

**Figure 3:** t* **fitting example.**

**The fitting maps of P-wave** t* **at different stations for the Yunnan Yiliang** $M_L$ **3.4 earthquake on January 6, 2013 (a-d), and the Lushan** $M_L$ **3.4 earthquake on January 12, 2016 (e-h).**

**t\* fitting results of the P-wave. In the upper subfigures, the black wavy lines represent waveforms at different stations from 5 s**
**before to 20 s after the original time, the blue wavy lines represent the P-wave signal windows, and the short black horizontal lines represent the the P-wave noise windows. In the lower subfigures, the blue curves represent the signal velocity spectra, the black dotted lines represent the noise velocity spectra, and the red curves represent the fitted curves. Letters from top to bottom represent station names, corner frequency, t\* values, and weights.**

Then we select the $V_P$ model of SWChinaCVM-1.0 constructed by Liu et al. (2021) to further invert the 3D $Q_P$ model of the
region by using the iterative least squares algorithm and SIMUL2000 program (Eberhart-Phillips, 1986; Thurber, 1993; Evans et al., 1994; Eberhart-Phillips and Michael, 1998).

$$t^* = \int_{ray\ path} \frac{1}{Q(s)*V(s)} ds \tag{2}$$

The initial average $Q_P$ value of 350 and the $Q_P$ damping value of 0.1 are selected. During the inversion, the weight of P-wave with a epicentral distance within 50 km is 1, the weight of P-wave with a epicentral distance between 50 and 200 km linearly
changes from 1 to 0, and the weight of P-wave above 200 km is 0. After 6 iterations, the data variance decreased by 46%.



## 3 Results

### 3.1 Checkerboard tests

We divide the study area into 0.25º × 0.25º grids with depth layers of 0, 5, 10, 15, 20, 25, and 30 km. The initial velocity values of the corresponding grid points are obtained by interpolation based on the SWChinaCVM-1.0. We will first evaluate

the resolution of the $Q_P$ model based on the checkerboard test method. Add 5% random noise to the data and add ±20% perturbation to the initial $Q$ value. The checkerboard test results show that the checkerboard with a grid of 50 km×50 km only recovers well at a depth of 10 km, with the $Q_P$ model in Changning area of Sichuan Basin and western Yunnan reaching a lateral resolution of 50 km at a depth of 10 km. The checkerboard test results of a 100 km×100 km grid (Fig. 4) show that the $Q_P$ model recovers well in the Sichuan Basin at a depth of 5 km, and overall recovers well in the study area at depths of

10 km and 15 km. The resolution is poor below 0 km and 20 km. The vertical profiles of $Q_P$ models along different latitudes (Fig. 5) show that the $Q_P$ model in Sichuan can recover well within a depth of 20 km, while the $Q_P$ model in Yunnan can recover well within a depth of 18 km. Therefore, the horizontal resolution of the 3D $Q_P$ model established in this paper for CSES is 100 km, and the vertical resolution is 5 km.







**Figure 4: Checkerboard test results of the layed $Q_P$ model.**

**The red solid line in (f) corresponds to each profile in Figure 5, and the letters above correspond to the numbers in Figure 5.**




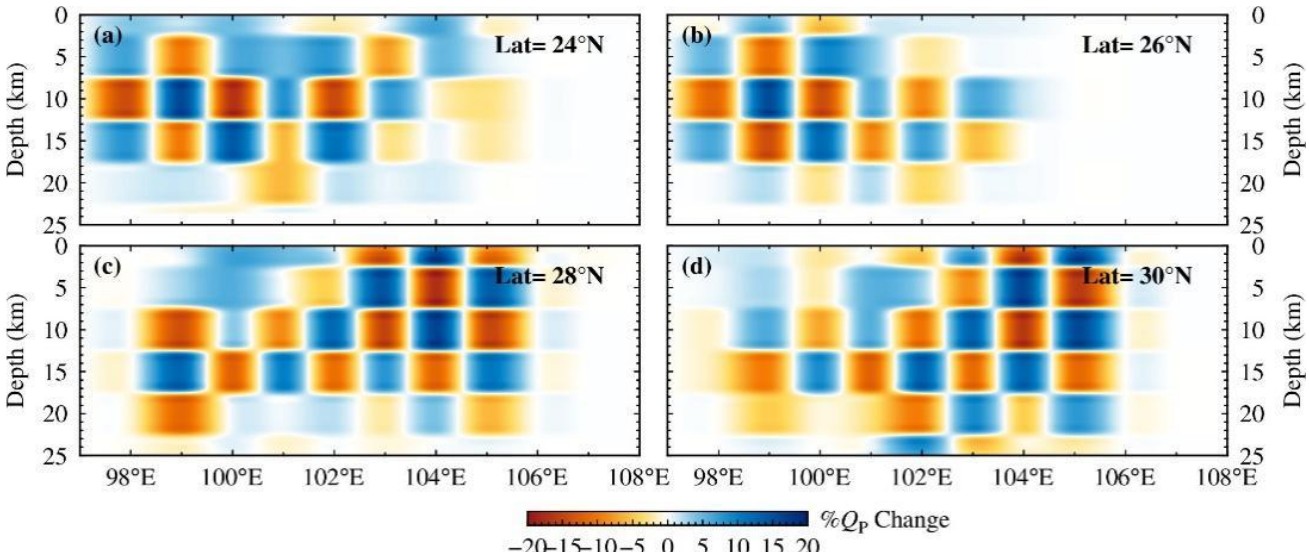

**Figure 5: Depth profile of $Q_P$ model along different latitudes in the checkerboard test.**

### 3.2 Resolution tests

Referring to the practices of previous researchers (Eberhart Phillips et al., 2008; Zhou et al., 2018; Duan et al., 2024), we further calculate the spread function (SF) values for each grid point from all the elements of the corresponding row of the resolution matrix to better evaluate the model resolution. Toomey and Foulger (1989) explain that the spread functions does not depend on grid spacing and damping. The quality of resolution is inversely proportional to the SF value, and it is generally used SF ≤ 4 as regions of acceptale $Q$ model quality (Eberhart Phillips and Michael, 1998). If SF > 4, it indicates

that the $Q$ model resolution is low and can only present rough features. We fix the $Q_P$ value of grid points with a number of rays less than 5 and obtain the layered SF image of the $Q_P$ model. The results show (Fig. 6) that, except for the Yangtze block, the SF values in most areas at depths of 5 km, 10 km and 15 km are less than 3, indicating that the Chuan-Dian region has better resolution at depths of 10 km and 15 km. At depths of 0 km, the southwestern Sichuan region has good resolution. At a depth of 20 km, the Sichuan Basin and western Yunnan region have good resolution. The SF profile of the $Q_P$ model

along the latitude direction (Fig. 7) shows that the resolution of the $Q_P$ model is relatively high in most areas of Sichuan and western Yunnan at depths of about 23 km.

    Integrating the results of the checkerboard test and the SF values for multiple resolution evaluations, we believe that the $Q$ model with SF < 4 in the study area is reliable. The SF ≥ 4 indicates little or no information, and the associated $Q$ model close to the initial model. The following will focus on the $Q_P$ model with SF < 4 within a depth range of 5-20 km.





**Figure 6: The spread function distribution of layered $Q_P$ model (SF < 4).**

**The black plus sign represents the grid points, and the gray filled circle represents the grid that does not participate in the inversion. (f) The red solid line corresponds to each profile in Figure 5, and the letters above correspond to the numbers in Figure 7.**




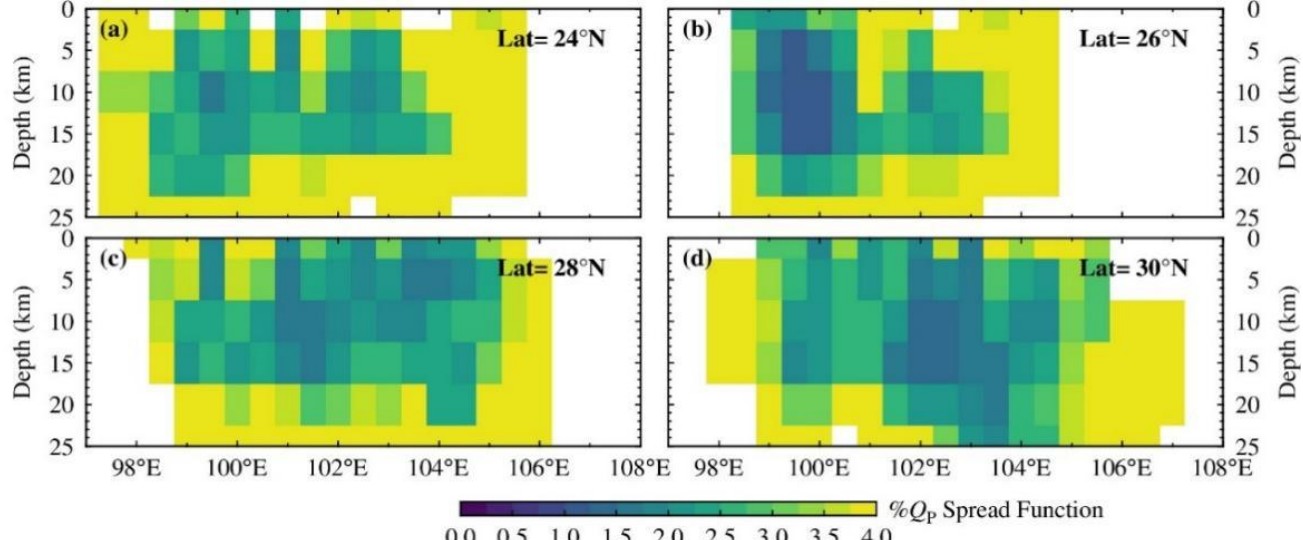


**Figure 7: The spread function distribution of the $Q_P$ model along different latitudes.**

### 3.3 3D $Q_P$ model in the CSES

The layered $Q_P$ model (Fig. 8) shows that the attenuation structure of CSES exhibits significant lateral heterogeneity. Along the main fault zones at depths of 0 km and 5 km are dominated by low $Q_P$ values (high attenuation), and there are high $Q_P$
anomalies (low attenuation) in the Songpan-Ganzi block (SPGZB) and the Chuan-Dian block (CDB). Hot springs are developed along the main fault zones (Fig. 8a), which correspond well with the shallow high attenuation anomalies, indicating that the shallow layers of the main fault zones are rich in fluids. The study area at a depth of 10 km is mainly characterized by high $Q_P$ anomalies, while low $Q_P$ anomalies are mainly distributed in the middle section of the Xiaojiang fault (XJF)  and the Simao basin. At a depth of 15 km, high $Q_P$ anomalies are predominant, distributed in the NS direction,
while low $Q_P$ anomalies are mainly found in the Sichuan basin (SCB) and areas such as Qujing and Simao. At a depth of 20 km, low $Q_P$ anomalies are predominant, while high $Q_P$ anomalies are distributed on both sides of the Longmenshan fault zone. At a depth of 25 km, high $Q_P$ anomalies still dominate along the Longmenshan fault zone.



**Figure 8: Layed $Q_P$ model (SF < 4).**

**(a) White diamonds represent the distribution of hot springs (Zhang et al., 2021). (b)-(e) Black stars indicate earthquakes of magnitude 6 or greater since 1970 within a 2.5 km depth range for each. The size of the symbols is proportional to the magnitude of the earthquakes. (f) Red solid lines represent the profiles, with letters denoting the names of the profiles.**

To better study the distribution characteristics of the medium structure in depth, we plot seven $Q_P$ profiles (Fig. 8f). At the same time, the $V_P$ model of SWChina CVM-1.0 and SWChina CVM-2.0 established by previous researchers are compared to

facilitate the comprehensive analysis of velocity and attenuation structure. Both the previous two versions of $V_P$ models and the $Q_P$ model obtained in this paper generally exhibit low-value anomalies in the shallow layers and high-value anomalies in



the deep layers. However, there are significant differences in the patterns of $Q_P$ anomalies obtained in this paper and $V_P$ anomalies from previous studies. In basin areas such as Sichuan and Simao basins, the middle and upper crust exhibits obvious low $Q_P$ anomalies (Figs. 9a, c). Within the CDB enclosed by multiple large active faults, both ends of NW and SE

exhibit low $Q_P$ characteristics throughout the upper and middle crust, while $V_P$ shows low-value anomalies only within 10 km (Fig. 9b). Beneath some large fault zones, such as the Jinshajiang fault (JSJF), Longmenshan fault (LMSF), and XJF, there are obvious low $Q_P$ anomalies in the entire upper and middle crust, with low $V_P$ anomalies distributed at depths within 10 km (Figs. 9e, g). There are also some large fault zones, such as the Huayingshan fault (HYSF), Red River fault (RRF), Daliangshan fault (DLSF), and the southern end of the Zemuhe fault (ZMHF), with low $V_P$ and low $Q_P$ anomalies at depths

within 10 km, and high $V_P$ and high $Q_P$ anomalies below 10 km (Figs. 9c, f, g). Notably, the bothe sides of the northern RRF and the junction of the southern segment of the RRF with the XJF, exhibit a clear low $Q_P$ anomaly within 10 km (Figs. 9c, g). In the Tengchong volcanic area, the upper and middle crust exhibit a low $Q_P$ anomaly that dips to the east (Fig. 9d), while $V_P$ shows a low value anomaly only within 5 km. In addition, there are high $Q_P$ anomalies below the Lancangjiang Fault (LCJF) (Figs. 9c, d), while Lianfeng fault (LFF) is mainly characterized by high $V_P$ and high $Q_P$ anomalies in the upper and middle

crust (Fig. 9f). The southern section of the Anninghe Fault (ANHF) exhibits high $Q_P$ anomalies (Figs. 9b, f). From the LMSF southward to the XJF, the upper crust shows high $V_P$ and high $Q_P$ anomalies (Fig. 9g). The differences in the distribution of low $Q_P$ anomalies and low $V_P$ anomalies in different regions obtained in this article indicate that the $Q_P$ model obtained in this paper can supplement the physical properties of the medium not revealed by the $V_P$ models. In the following, we will analyze and discuss the medium's properties and the seismogenic environment for moderate and strong earthquakes by

integrating the characteristics of $V_P$ and $Q_P$ in CSES.





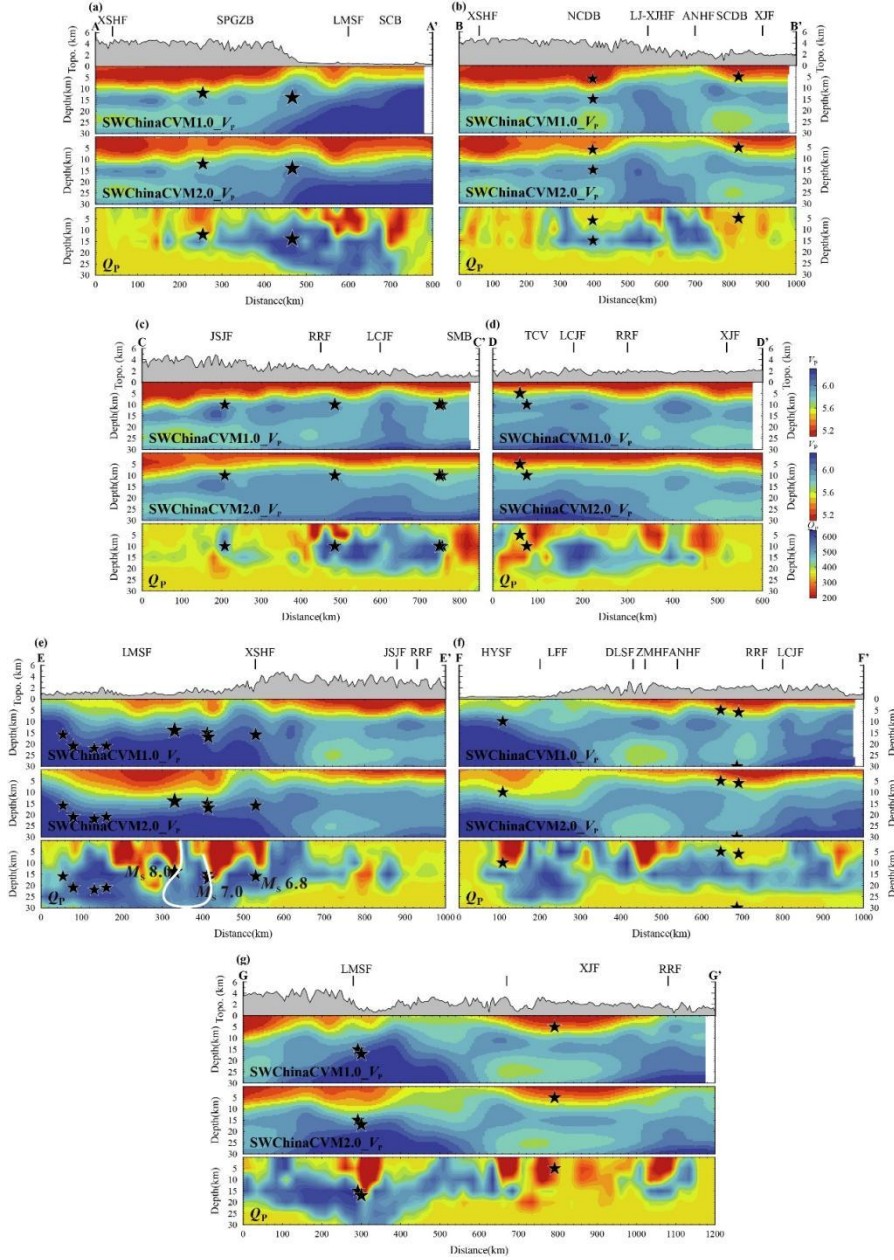

**Figure 9: $V_P$ and $Q_P$ models of various profiles.**

Each subplot from top to bottom represents the $V_P$ results of SWChinaCVM1.0 and 2.0, and the $Q_P$ results obtained in this paper. The black bar above the topographic map represents the position of the fault. Black stars represent earthquakes with M ≥ 6 within 25 km on both sides of the profile since 1970. The size of the symbols is proportional to the magnitude of the earthquakes. The white curve in (e) encloses the low-attenuation area between the 2008 Wenchuan $M_S$ 8.0 earthquake and the 2013 Lushan $M_S$ 7.0 earthquake. Other symbols are the same as in Figure 8.



## 4 Discussion

### 4.1 Spatial distribution characteristics of media structure in typical structural areas

The $V_P$ models from SWChinaCVM-1.0 and SWChinaCVM-2.0 indicate, with the LMSF as the boundary, the depth of the low $V_P$ anomaly in the SPGZB on the west side is greater than that to in the SCB on the east side (Fig. 9a). In contrast, the $Q_P$ model shows that within 10 km of the shallow layer, the SPGZB exhibits low attenuation anomalies while the SCB shows high attenuation anomalies (Fig. 9a), which is consistent with the $V_S$ model and the thickness of the sedimentary layers obtained by Yang et al. (2023), as well as the $V_P$ and $V_S$ models obtained by Wu et al. (2024). We believe that the distinct

low $V_P$, low $V_S$, and low $Q_P$ anomalies in the shallow layers of the SCB reflect the presence of several kilometers of Cenozoic-Mesozoic sedimentary rocks in the SCB.

    The NW-trending BB' profile across the CDB reveals that the northwestern and southeastern segments of the CDB exhibit high attenuation characteristics in the upper and middle crust, while the central region is predominantly characterized by low attenuation features. The low attenuation feature between the Lijiang-Xiaojinhe fault (LJ-XJHF) and ANHF corresponds to

the high velocity anomalies obtained by Wu et al. (2024) along the similar profile. The BB' profile crosses the inner zone of the Emeishan Large Igneous Province (ELIP), which is primarily composed of flood basalts, accompanied by mafic and felsic intrusive rocks (Ren et al., 2022). Various geophysical studies have indicated that the inner zone of the ELIP is a rigid region with high $V_P$, high $V_S$, high Poisson's ratio, high density, and high resistivity (Bao et al., 2015; Xu et al., 2015; Li et al., 2020; Zhang et al., 2020), which coincides with the high $Q_P$ anomalies obtained in this study in the CDB. Therefore, the high

$Q_P$ anomalies reflect the basalt characteristics of the inner zone of the ELIP. At a depth of 20 km, the high attenuation anomalies of the CDB are connected near Xichang (Fig. 8e), which is similar to the 2 Hz $Q_{Lg}$ model obtained by Zhou et al. (2008) in the Sichuan-Yunnan region. However, it differs from the $Q_{Lg}$ model (0.2-2.0 Hz) in the Tibetan Plateau obtained by Zhao et al. (2013). Previous velocity models have also shown (Yao et al., 2008; Bao et al., 2015; Yang et al., 2020, 2023; Liu et al., 2021, 2023; Wu et al., 2024) that the low-velocity zone located in the northwestern CDB and the low-velocity

zone along the Xiaojiang Fault are separated by the ELIP. Among them, the $V_S$ model obtained by Yang et al. (2023) shows that in the middle and lower crust at a depth of 20-30 km, the low-velocity anomalies of the western Sichuan and the XJF are significantly separated by a high-velocity body, which extends continuously from the east side of the SCB, crosses the ANHF, and terminates at the central Yunnan block. Since the $Q_P$ model obtained in this study in the CDB have no resolution below 20 km, it is not possible to determine the characteristics of $Q_P$ in the middle and lower crust of the CDB, and it is not

possible to accurately describe the distribution of flow in the middle and lower crust of the Tibetan Plateau.

    Significant low-velocity and high-attenuation anomalies are observed in the shallow layers along LMSF (Fig. 9e), and the range of high attenuation anomalies is deeper. Between the LMSF and the XSHF, two distinct high-attenuation zones are present in the upper crust, corresponding to the 2008 Wenchuan $M_S$ 8.0 earthquake and the 2013 Lushan $M_S$ 7.0 earthquake source areas, respectively. The high-attenuation anomaly above the epicenter of the Wenchuan $M_S$ 8.0 earthquake is

consistent with the $Q_P$ and $Q_S$ models obtained by Zhou (2016) and the low-resistivity anomaly characteristics reported by



Zhao et al. (2012). Zhao et al. (2012) suggested that the high-conductivity body beneath the LMSF in the Wenchuan earthquake source area may reflect an increase in fluid content. We believe that the two high-attenuation zones in the shallow layers of the LMSF indicate the presence of mechanically weak zones along the fault. Considering the development of hot springs along the LMSF, these mechanically weak zones are likely influenced by fluids in the upper crust. The occurrence of the Wenchuan $M_S$ 8.0 earthquake and the Lushan $M_S$ 7.0 earthquake may be related to the role of fluids.

The Xiaojiang fault (XJF) is the southeastern boundary of the CDB, extends over 400 km from north of Qiaojia to southeast of Jianshui. The fault zone is situated in a high heat flow area of 85 mW/m2 (Yuan et al., 2006), along which  more than 20 basins with different scales, and hot springs are developed. The north-south trending GG' profile reveals (Fig. 9g) that the entire upper and middle crust along the XJF is dominated by significant low $Q_P$ anomalies, which terminate at the southern end of the Red River fault (RRF). These low $Q_P$ anomalies are in good agreement with the $Q_{Lg}$ models at different frequencies (Zhou et al., 2008; Zhao et al., 2013) and the $Q_P$ model obtained by previous studies (Dai et al., 2020). The resolution  of the $Q_P$ model along the XJF in this study is superior to that of Dai et al. (2020) at depths of 5 km and 15 km. The low $Q_P$ anomalies obtained in this study also show good consistency with the low $V_P$ anomalies reported by previous studies (Wu et al., 2013, 2024). The electrical resistivity structure along the similar profile shows an alternating pattern of high and low anomalies (Yu et al., 2022), and the low resistivity anomalies in the upper and middle crust may reflect the presence of saline fluids and highly conductive minerals. Geochemical studies also indicate that the XJF is dominated by crustal heat flow (Zhang et al., 2021). Therefore, the low-velocity and high-attenuation anomalies in the upper and middle crust of the XJF reflect abundant fluids in the fault zone. The prominent high-attenuation anomalies are concentrated at the northern and southern ends of the XJF. Geochemical research also shows that the relatively high temperatures and ion concentrations of hot spring water, as well as the relatively high fluxes of soil gas radon and carbon dioxide, are concentrated in the northern and southern segments of the XJF, which corresponds to the spatial distribution of seismicity and fault slip rates (He et al., 2023). Shi and Wang (2017) shows that the permeability is higher in the northern and southern segments of the XJF. In summary, we suggest that the low-velocity, low-resistivity, and high-attenuation anomalies in the upper crust of the XJF reflect the presence of fluids and highly conductive minerals beneath the fault zone. Intense fault activity in the southern and northern segments leads to high degree of crustal medium fragmentation, which may also enhance permeability and accelerate the water-rock interaction and soil gas emission within the fault zone.

### 4.2 Low $Q_P$ anomaly in the Western Yunnan region

The CC' profile along the Jinsha River fault (JSJF) shows that (Fig. 9c) the fault is predominantly characterized by low $Q_P$ anomalies, and high $Q_P$ anomalies are dominant in the south of the Red River fault (RRF). Zhou et al. (2020) found that the He in the hot spring gases in the JSJF-RRF is mainly from the crust, and the high values of H2 concentration and He isotopic ratios appear at three fracture junctions of the JSJF with the Batang fault, the Zhongdian fault with the RRF, and the southern section of RRF with the XJF, respectively, which has a good corresponding relationship with the high attenuation anomalies obtained in different depths in this paper, especially in the intersection area of Zhongdian fault and RRF (Fig. 9c) and the



intersection area of the southern section of RRF and XJF (Fig. 9g). The Simao basin exhibits low $Q_P$ anomalies. It is a
Mesozoic Cenozoic sedimentary basin, mainly composed of mudstone and sandstone. There are many hot springs and salt
springs in the basin. The obvious high attenuation anomalies in the basin reflect that it being fluid- rich in the upper crust.

The Tengchong volcano (TCV), located in the southeastern Tibet Plateau, is one of the largest active volcanoes in China (Fig.
1a). TCV is characterized by a large amount of magmatic gas (carbon dioxide and sulfide) emissions (Zhao et al., 2011,
2012), active hydrothermal cycle (Jiang et al., 2019), high surface heat flow (≥ 90 mW/m2), and strong earthquakes during
the Holocene (Zhao et al., 2020) and volcanic activity (Wang et al., 2007; Zou et al., 2014). The measurement of hot spring
fluids also shows that there are a large number of hot spring fluids in the TCV (Fig. 8a), indicating that the TCV is rich in
thermal materials and fluids. Previous inversion of the crust beneath Tengchong volcano yielded low $V_S$ anomalies (Shen et
al., 2022; Yang et al., 2023; Lin et al., 2024). The DD' profile across the TCV shows a high-attenuation anomaly that dips to
the west beneath the TCV (Fig. 9d). Zhao et al. (2021) found the low $V_S$ anomaly at the depth of 20-35 km to the west of the
TCV, which revealed a large basaltic magma reservoir and the melt fraction of 2% -4.5%. The high attenuation anomaly
obtained in the middle and upper crust of TCV in this paper is likely connected to the low $V_S$ anomaly obtained in the middle
and lower crust by previous researchers. The westard- dipping high attenuation anomaly in the upper and middle crust of the
TCV depicts the possible upwelling of deep magma from west to east, which is also a direct reflection of partial melting and
fluids.

**4.3 Seismogenic Mechanism of Moderate and Strong Earthquakes**

Most of the historical earthquakes in Fig. 1 lacked depth information before 1970. Therefore, we only consider the
relationship between earthquakes with M ≥ 6 and the $Q_P$ model in CSES after 1970. Some studies have found that large
crustal earthquakes often occur in high-velocity anomaly areas (Liu et al., 2023; Huang and Zhao, 2004; Pei et al., 2019; Sun
et al., 2021). They believe that the high-velocity anomaly area may be the asperity of the fault, with high stress
accumulation, resulting in large earthquakes. Other studies have found that moderate and strong earthquakes with M ≥ 6
often occur in high attenuation areas or at the boundaries of high and low attenuation (Zhou et al., 2008; Pei et al., 2009; Liu
and Zhao, 2015; Zhou et al., 2020). These results support that fluids increase the pore pressure of faults and promote
earthquake nucleation. In this paper, the earthquakes within 2.5 km of each depth layer are projected onto the layered $Q_P$
model, and the earthquakes with M ≥ 6 within 25 km on both sides of each profile are projected onto the $Q_P$ model (Figs. 8,
9). The results show that, compared with the velocity models, the relationship between earthquakes above M 6 and the
attenuation model is closer (Fig. 9). Earthquakes with M ≥ 6 in western Yunnan are distributed in the low-attenuation area
and the boundary of high and low attenuation anomalies (Fig. 9c). Two major earthquakes in TVC occurred in the high
attenuation area (Fig. 9d). Earthquakes with M ≥ 6 along the northeastern segment of the Longmenshan fault (LMSF) are
aftershocks of the 2008 Wenchuan $M_S$ 8.0 earthquake, which all occurred in the low attenuation area or the boundary of high
and low attenuation (Fig. 9e). The major earthquakes in southwest section of the LMSF are 2008 Wenchuan $M_S$ 8.0
earthquake, 2013 Lushan $M_S$ 7.0 earthquake, 2022 Lushan $M_S$ 6.1 earthquake and Luding $M_S$ 6.8 earthquake from northeast



to southwest, which all occurred at the boundary of high-low attenuation anomalies or low attenuation area (Figs. 9e, g), similar to the $V_P$ imaging results obtained by previous studies (Li et al., 2013; Pei et al., 2014; Liu et al., 2023). Earthquakes with M ≥ 6 along the Huayingshan fault occurred at the boundary of high and low attenuation (Fig. 9f). Our result shows that

earthquakes with M ≥ 6 in CSES generally occur in the low attenuation area or the boundary area of high and low attenuation anomalies. The low attenuation zone may represent the asperity with high mechanical strength, which is conducive to stress accumulation. The boundary of high and low attenuation anomalies corresponds to the gradient zone of stress change, which is more prone to sudden change in stress. Both cases promote the rupture of faults and trigger large earthquakes.

Our result also shows that the source area of the Lushan earthquake and the Wenchuan earthquake are separated by a low attenuation zone (Fig. 9e). The results of body wave velocity imaging showed that the Wenchuan-Lushan seismic gap is located in the low-velocity anomaly area (Li et al., 2013; Pei et al., 2014). They believed that the intensity of this area is low and not enough to generate strong earthquakes. Wang et al. (2015) observed an obvious seismic moment deficit in the seismic gap through GPS velocity data, suggesting its potential to prepare strong earthquakes. Wang et al. (2018) found that

the microseismic activity in the seismic gap is relatively weak compare to the north and south sides, indicating that the possibility of accumulated stress and strain being released through microseisms is very small. Diao et al. (2018) studied the post earthquake mechanism of the Wenchuan earthquake and found that there is almost no afterslip distribution in the gap, and it is still accumulating strain through GPS observation, suggesting that the gap has a relatively large seismic risk. In addition, based on the study of Coulomb stress, it was found that the Wenchuan earthquake and Lushan earthquake have

greatly enhanced the stress in the gap, thus greatly increasing the possibility of strong earthquakes (Guo et al., 2020). The low attenuation anomaly region in the seismic gap between the two strong earthquake obtained by this paper means that the medium strength in this region is high and is still in the stress accumulation. Contrary to the low-velocity characteristics, it is consistent with the observation results of GPS. Therefore, we support that the seismic gap poses a danger of major earthquakes in the future.

**5 Conclusions**

In this study, we collect and sort out the earthquake catalog, phase reports and seismic waveforms of magnitude 1.5 or above recorded by 582 stations in the China Seismic Experimental Site from 2013 to 2023. Through seismic clustering and phase quality control, we select 35, 778 high-quality seismic waveforms. The high resolution 3D $Q_P$ model of CSES is obtained by using SIMUL2000 program. The horizontal resolution of the model is 50 km and the vertical resolution is 5 km. Combined

with other geophysical inversion results, geochemical observation and geological structure data, this paper has the following understanding of the medium environment of the middle and upper crust of CSES:

(1) In the depth of 5 km along the large fault zone in the Chuan-Dian region, the $Q_P$ model exhibits low-value anomalies, which corresponds well with the characteristics of hot spring development around the fault zone, reflecting the strong



tectonic movement of major active fault zones leading to a highly fragmented and fluid-rich shallow medium. The $Q_P$ model

in the upper and middle crust of Sichuan Basin and Simao basin also shows low-value anomalies, reflecting the characteristics of thicker sedimentary layers in these areas.

(2) The high attenuation anomalies obtained in the upper and middle crust of Tengchong volcano is likely to be connected with the low $V_S$ anomaly in the middle and lower crust. The westward-dipping high attenuation anomalies in the upper and middle crust depict the upwelling the pattern of magma uprising from west to east.

(3) There are two obvious high attenuation anomaly areas under the Longmenshan fault, which are located above the epicenter of the 2008 Wenchuan $M_S$ 8.0 earthquake and 2013 Lushan $M_S$ 7.0 earthquake, respectively. The fluids in the upper crust may promote the nucleation of the two large earthquakes. The two seismic source areas are separated by a low attenuation area, which may has high stress accumulation and still have the risk of large earthquakes in the future.

(4) The Xiaojiang fault shows high attenuation anomalies in the middle and upper crust, and the anomalies at the north and

south ends are more obvious. The high attenuation anomaly at the intersection of the southern end and the Red River fault corresponds well to the high H2 and He isotopic content in hot springs, indicating that the Xiaojiang fault is rich in fluid and high conductivity minerals in upper and middle crust. The strong fault activity in the southern and northern segments leads to high degree of crustal media fragmentation and enhances the permeability of the fault zone.

(5) Most moderate and strong earthquakes with M ≥ 6 in CSES occur in the low attenuation areas or the boundary area of

high-low attenuation anomalies, which is similar to the spatial relationship between earthquakes and velocity anomalies. This paper considers that such an abnormal area is conducive to stress accumulation and sudden change in stress, which is prone to large earthquakes.

**Code/Data Availability**

The earthquake catalogues, phase reports are provided by China Earthquake Networks Center, National Earthquake Data

Center (CENC, 2021), Sichuan Earthquake Administration and Yunnan Earthquake Administration. The inverted 3D $Q_P$ models are available at doi: 10.5281/zenodo.13994425. The version v1 of the Simul2000 program used for 3D seismic tomography is preserved at doi:10.5281/zenodo.5547888, available via Thurber & Eberhart-Phillips (2021). All figures are made with the Generic Mapping Tools Version 6 (Wessel et al., 2019) and CorelDRAW 2020 (Copyright © 2020 Corel Corporation) from https://www. corel.com/.

**Competing Interests**

The authors acknowledge that there are no conflicts of interest.



## Acknowledgements

This study was jointly funded by the National Key R&D Program (2021YFC3000704), the National Natural Science Foundation of China (42174066) and the Central Public-interest Scientific Institution Basal Research Fund
(CEAIEF20240405). Most of figures in this paper were plotted by GMT 6 software (Wessel et al., 2019).

## Author Contribution

Lianqing Zhou wrote, reviewed and edited the manuscript.

Mengqiao Duan wrote the original manuscript.

Ying Fu provided data for Sichuan Province.

Yanru An supplemented the data for Sichuan and Yunnan provinces.

Jingqiong Yang provided data for Yunnan Province.

Lianqing Zhou and Xiaodong Zhang provided project funding for this research.

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
