# Peer review of "The 3D $Q_P$ Model of the China Seismic Experiment Site (CSES-Q1.0) and Its Tectonic Implications"

_EGUsphere, 2024_

## Author Response (AR1)

**Response to editor**

Dear Lianqing Zhou,

We are pleased to inform you that the validation of your following SE submission through EGUsphere is finished: egusphere-2024-3340

Title: The 3D Qp Model of the China Seismic Experiment Site (CSES-Q1.0) and Its Tectonic Implications

Author(s): Mengqiao Duan et al.

MS type: Research article

Iteration: Initial submission

Special issue: Seismic imaging from the lithosphere to the near surface

Currently, we are asking the topic editors to handle your manuscript and you will be informed about the outcome by separate email.

For now, we will proceed with your manuscript as submitted. However, please adjust your manuscript files before your next file upload (next round of revision or after acceptance) considering the following requirements:

Please note that your reference list has not been compiled according to our standards. Please consider adjusting your reference list with the next revision of your manuscript. The manuscript preparation guidelines can be seen at: https://www.solid-earth.net/for_authors/manuscript_preparation.html.

R. We has corrected the reference list. Please see the reference section in the revised manuscript.

A "Short summary" system section contains scientific abbreviations.

Please note, if you used scientific abbreviations without giving the written-out explanation, these must be written out with the next file upload request. However, do not forget that there is a limit to characters (not words!) for "Short summary": it must be < 500 characters.

R. We have supplemented "Short summary".

Please log in with your Copernicus Office user ID 717319 to monitor the processing of your manuscript via your MS overview

at: https://editor.copernicus.org/EGUsphere/my_manuscript_overview.

In case any questions arise, please do not hesitate to contact me. Thank you very much.

Kind regards,

The editorial support team

Copernicus Publications

editorial@copernicus.org

**Response to reviews**

Below is point-by-point response. The line numbers refer to the approximate location in the highlighted copy. All the original review text is kept for the convenience of tracking.

**Review Comments**

**Reviewer #1**

This paper presents attenuation tomography of southwestern China using a large dataset, yielding an updated, high-resolution attenuation model that complements existing velocity models. The new model provides valuable insights into fault zone structures and seismic risk in the region. Overall, the manuscript is well written, and both the methodology and inversion results appear robust. My primary concerns pertain to the need for additional technical details and consideration of alternative explanations for the high-attenuation fault zones. These suggestions do not require new tests or experiments and can likely be addressed with minor revisions. Please see the detailed comments below:

1. While Q tomography is a well-established method, a more comprehensive description of the method would benefit this paper. In particular, please elaborate on

the spectrum-fitting algorithm, the kernel construction, and the formulation of the inverse problem to ensure readers can fully understand your approach.

R. The algorithm and formulas for iteratively fitting the $\Omega_{0ij}$ and $t^*$ are as follows:.

The velocity amplitude spectrum of the $i$-th earthquake recorded by the $j$-th station.

$$A_{ij}(f) = 2\pi f \cdot \frac{\Omega_{0ij} \cdot f_{ci}^2}{f_{ci}^2 + f^2} e^{-\pi f t_{ij}^*} \tag{1}$$

in which f is the frequency, $f_{ci}$ is the corner frequency of the event i, $\Omega_{0ij}$ is the spectral level at low frequency of the record ij, $t_{ij}^*$ is the whole path attenuation term $t/Q$ of the record ij.

The $A_{ij}(f)$ can be calculated from seismic waveforms. The unknown parameters $\Omega_{0ij}$, $f_{ci}$ , and $t_{ij}^*$ can be solved using an iterative damped least squares inversion method. The iterative steps are outlined as follows:

   (1) First, determine the fitting frequency band, such as 2-20 Hz.
   (2) Then, select valid records based on the signal-to-noise ratio.
   (3) For all records of an event, use a grid search method within a certain frequency range to solve for the corner frequency of each event $f_{ci}$. Specifically, from 2 Hz to 20 Hz at intervals of 0.1 Hz, continuously increase the value of $f_{ci}$ and calculate the theoretical velocity spectrum $D_{ij}(f)$. Assume an initial value of $t_{ij}^*$ is 0.02, fix $f_{ci}$ and an estimated value $t_{ij}^*$, and use the following formula to calculate the zero-frequency spectral value $\Omega_{0ij}$:

$$\Omega_{0ij} = \frac{\sum_{f<f_{ci}} D_{ij}(f) * A_{ij}(f)}{\sum_{f<f_{ci}} A_{ij}(f) * A_{ij}(f)} \tag{2}$$

In which $A_{ij}(f)$ is the theoretical velocity spectrum,$D_{ij}(f)$ is the observed velocity spectrum.

   (4) By fixing $\Omega_{0ij}$ and $f_{ci}$, a better $t_{ij}^*$ can be obtained.

$$t_{ij}^* = \frac{\sum \log(A_{ij}(f)) * f - \sum \log(D_{ij}(f)) * f}{\pi \sum f * f} \tag{3}$$

Repeating steps (3) and (4) for *n* iterations, each iteration obtaining a new $t_{ij}^*$ and recalculating $\Omega_{0ij}$. The corner frequency $f_{ci}$ for the event i is determined by the frequency that minimizes the fitting error between the theoretical and the observed velocity spectrum. The expression for the fitting error is:

$$fit = \frac{1}{N}\sum_{n=1}^{N} \log\left[A_{ij}(f) - D_{ij}(f)\right]^2 \tag{4}$$

Where $N$ is the number of sample points in the range of 2-20 Hz for each station of each event. When the sum fitting errors of all stations is minimum, the corresponding grid search frequency is the corner frequency of the event *i*. $t_{ij}^*$ and $\Omega_{0ij}$ can be calculated by the final $f_{ci}$.

It should be noted that in the original text, "the fit values with the fitting error less than 0.1 s, 0.2 s, 0.3 s and 0.4 s" should be changed to "0.1, 0.2, 0.3, 0.4," without units. We have incorporated the above explanations into the revised manuscript. **Please see "2.2 Method" section in the revised manuscript.**

2. From my own experience with Lg-wave attenuation, Q measurements can be sensitive to the frequency spectrum characteristics (e.g., spectrum holes) and the chosen fitting algorithm. Hence, some quality control may be required to preserve reliable measurements. I suggest showing the distribution of t* measurements as a function of distance, which may help to detect any potential systematic patterns or outliers.

R. We agree with your opinion, Q value does have frequency dependence. We assume frequency independent attenuation in this paper. Studies that solve for 3-D Q with a range of α find that the resulting $Q$ models are similar and equivalent in terms of interpreting $Q$ structure (Lees & Lindley, 1994). Eberhart-Phillips & Chadwick (2002) believed that over the typical usable frequency range of their observations a frequency dependence of 0.3 would make < 10% difference in the amplitude decay compared to frequency independence. Eberhart-Phillips et al. (2014) tested different frequency-dependent factors (α = 0.1-0.6), they found that most spectra have equivalent fit with or without frequency dependence, and a small proportion favor

either α = 0 or α = 0.4-0.6. Thus, they cannot determine that frequency dependence is necessary. Because of most other studies believe that α = 0.5, and thus they use 0.5 for comparison of frequency-dependent $Q$ and found a strong linear relation. Hence 3-D $Q$ models would have similar patterns for α = 0 and α = 0.5, and they choose to obtain frequency independent 3-D $Q$ models. Therefore, we believe that the frequency independence $Q$ values may be higher, but the medium characteristics revealed by it will be not change much.

To obtain high-quality $t*$ data, we conducted quality control in the following ways:

1. **Testing the fitting frequency band**: By testing different fitting frequency bands, we ultimately selected the 2-20 Hz band as the $t^*$ fitting frequency band to obtain more high-quality $t^*$ measurements.

2. **Weighting $t^*$ based on fitting error**: We discarded $t^*$ data with fitting errors greater than 0.4 while ensuring that each event had at least three $t^*$ data points.

3. **Removing unreliable $t^*$ data**: During the inversion of $Q$ values, the program automatically discarded unreliable $t^*$ data with $Q>1500$.

4. **Assigning different weights to epicentral distances**: During inversion, we weighted $t^*$ data from different epicentral distances to further leverage the role of near-field stations in the inversion process.

5. **Testing the impact of velocity models**: Our previous studies have also discussed that the impact of velocity models on $Q$ value inversion is minimal (Duan et al., 2024).

Following your suggestion, we plotted the relationship between $t^*$ and epicentral distance (Figure R1) and found no significant correlation between the two. Some $t^*$ values close to zero were automatically excluded by the program during inversion.

[Figure]

Figure R1 The relationship between epicentral distance and $t^*$.

3. For the low-Q anomalies along major fault zones (e.g., LMSF, XJF), fluid-related attenuation is undoubtedly a key factor. However, scattering attenuation can also contribute to seismic energy loss, especially in complex, fragmented fault zones following large earthquakes. Please consider exploring scattering as an alternative or additional explanation for these observations.

R. Thank you for your suggestion. The main rupture of the Wenchuan earthquake occurred near Yingxiu on the Beichuan Fault, with both the Beichuan Fault and the Pengguan Fault experiencing severe ruptures, with rupture lengths reaching 240–300 km and ~90 km, respectively (Zhang et al., 2009a). These two major earthquakes have significantly fragmented the medium along the Longmenshan Fault. Additionally, a series of complex rock bodies (757–805 Ma) are exposed along the Longmenshan Fault from north to south, including the Nanba Complex, Pengguan Complex, Baoxing Complex, and Kangding Complex (Zhang et al., 2009b). Therefore, the high attenuation anomaly may partly reflect strong medium inhomogeneity, leading to scattering attenuation. XJF has experienced multiple tectonic movements, with acidic and basic magmatic intrusions during the Jinning, Caledonian, and Hercynian periods,

causing large amounts of basalt and ultrabasic rock bodies to be exposed along the fault (Li, 1993). The high attenuation observed along the fault may also be due to scattering attenuation caused by the inhomogeneity of these unconsolidated rocks.

We added some explanation for scattering attenuation along LMSF and XJF. **Please see "4.1 Spatial distribution characteristics of media structure in typical structural areas" section in the revised manuscript.**

**Minor Comments**

Figure 2: Consider using density plots to illustrate the distribution of earthquakes before and after declustering.

R. Corrected. **Please see Figure 2 in the revised manuscript.**

Line 119: Please clarify why a threshold value of 2.56 was selected.

R. We have referenced several studies. They show that traditional methods of t*estimation often used the fixed time window such as 2.56s after the P-wave onset to calculate the observed velocity spectra of P waves and fit t* (Lees & Lindley1994; Eberhart-Phillips & Chadwick 2002; Hauksson & Shearer, 2006). Some earthquakes have epicentral distances of less than 20 km, and the S-P traveltime differences are less than 2.56 s. For these closer stations, the window length for the recorded P-wave is set to the S-P traveltime differences. **Please see lines 118-123 in the revised manuscript.**

Line 124: Provide more details on the "iterative algorithm" used for spectrum fitting.

R. Please refer to the response to the first question. **Please see"2.2 Method"section in the revised manuscript.**

Line 145: Explain how the weight varies with epicentral distance and, if possible, include a comparison of the data misfit distribution between the initial and final models.

R. During the inversion, t* data with hypocentral distances less than 50 km were weighted by a factor of 1, and t* data with hypocentral distances between 50 and 200 km were weighted by a linear function between 1 and 0, and t* data with hypocentral distances larger than 200 km were discarded. After 6 iterations, the final data variance of $Q_P$ inversion decreased from 0.00054 to 0.000294, a reducation of 46% compared with the first inversion. **Please see lines 161-164 in the revised manuscript.**

Line 150: Change "Add 5%" to "We add 5%."

**R. Corrected. Please see line 175 in the revised manuscript.**

Figure 4: Show the major fault zones on the checkerboard plot as well, and consider adding markers (e.g., circles) for key cities/towns.

R. Corrected. **Please see Figure 4 in the revised manuscript.**

Lines 166–167: The resolution matrix has not been introduced in previous sections. Please add an explanation of the inversion process and the role of the resolution matrix in the methodology section.

R: We have added the explanation in the manuscript. The revised inversion process also described in the Methods section. **Please see lines 164-168 in the revised manuscript.**

Line 170: Correct the typo "fixe" to "fix."

R: Corrected. **Please see line 197 in the revised manuscript.**

Line 239: Replace "Wu et al. (2024)" with the model name "CSES-VM1.0."

R. Corrected. **Please see lines 266 and 272 in the revised manuscript.**

Line 272: For heat flow units (mW/m²), ensure the square is properly subscripted.

R. The heat flow units are right.

Line 344: Change "prepare strong earthquakes" to "host strong earthquakes."

R. Corrected. **Please see line 378 in the revised manuscript.**

Line 351: Change "means" to "suggests."

R. Corrected. **Please see line 385 in the revised manuscript.**

Citation: https://doi.org/10.5194/egusphere-2024-3340-RC1

**Reviewer #2**

The manuscript presents a high-resolution 3D Qp model for the China Seismic Experiment Site (CSES) and provides valuable insights into the tectonic implications of attenuation structures. The study is methodologically sound, well-structured, and addresses a critical gap in seismic attenuation modeling for this seismically active region. The integration of velocity models, geochemical observations, and geological data strengthens the conclusions. I think this is a good study and their interpretations are reasonable to me. As far as I understand, the model proposed in this article is currently the first high-resolution attenuation model in the Sichuan-Yunnan region and is recommended as a public model for the China Earthquake Science Experiment Field. The manuscript is otherwise well-prepared and merits publication in Solid Earth after minor revisions.

Minor Revisions:

1. It is more appropriate to change CSES-Q1.0 to CSESQ-V1.0 in the title.

R. Thank you for your suggestions. Considering that CSES is a proprietary abbreviation for the China Seismic Experimental Site, combining it with $Q$ might not be easily recognizable. Since V can be easily misinterpreted as Velocity, to avoid confusion, we prefer to retain the abbreviation **CSES-Q1.0**.

2. Minor grammatical inconsistencies exist (e.g., "layed" in Figure 4 caption should be "layered"). A thorough proofread is recommended.

R. Corrected. **Please see captions of figures 4 and 8 in the revised manuscript.**

3. L97: Clarify earthquakes with at least six phases" means six P phases or both P and S phases?

R. It means six both P and S phases. **Please see line 95 in the revised manuscript.**

4. Please provide additional details of SIMUL2000 parameters on damping factor and weight setting in section 2.2.

R. We set maximum number of iterations to 10 to perform inversions with $Q$ values between 50 and 650. According to the minimum value of data variance, the initial values of $Q_P$ was set to 350 with 9 iterations. Then, according to the trade-off curve between the model variance and the data variance from the inversion with a single iteration, the optimal damping values of $Q_P$ tomography was set to 0.1. We have added the details in the manuscript. **Please see lines 157-160 in the revised manuscript.**

5. Standardize journal abbreviations (e.g., "Geophys. J. Int." vs. "Geophys J Int", "J. Geophys. Res. Solid Earth" vs "J. Geophys. Res.-Solid Earth" and "J. Geophys. Res.").

R. Corrected. **Please see the References in the revised manuscript.**

6. Figure 2 appears relatively blurry, and the distinction between earthquakes in (a) and (b) is not very clear. It is recommended to represent the earthquakes with hollow circles or solid circles with borders, and upload a vector image.

R. In accordance with the suggestions of Reviewer 1, we use density plots to illustrate the distribution of earthquakes before and after declustering. **Please see Figure 2 in the revised manuscript and we have uploaded the vector images.**

7. Figure 3 and 4 are also not very clear, please upload vector images.

R. We have uploaded the vector images of the two figures.

8. The labels of the legend in Figure 5 are not seperated, please correct them.

R. Corrected. **Please see figure 5 in the revised manuscript.**

9. Change "Spread_Fuction" to "Spread Function" in Figure 6.

R. Corrected. Please see**figure 6 in the revised manuscript.**

**Citation**: https://doi.org/10.5194/egusphere-2024-3340-RC2

**References**

Duan, M., L. Zhou, C. Zhao, Z. Liu, and X. Zhang (2024). High-Resolution 3D QP and QS Models of the Middle Eastern Boundary of the Sichuan–Yunnan Rhombic Block: New Insight into Implication for Seismogenesis, *Seismol. Res. Lett.*, doi: 10.1785/0220230232.

Hauksson, E., and P. M. Shearer (2006). Attenuation models (Qp and Qs) in three dimensions of the southern California crust : Inferred fluid saturation at seismogenic depths, *J. Geophys. Res.-Solid Earth* **111**, no. B5, B05302–B05322, doi: 10.1029/2005JB003947.

Lees, J. M., and G. T. Lindley (1994). Three-dimensional attenuation tomography at Loma Prieta: Inversion of t* for Q, *J. Geophys. Res.-Solid Earth* **99**, no. B4, 6843–6863, doi: 10.1029/93JB03460.

Li, P. (1993). The Xianshuihe-Xiaojiang fault zone [*M*]. Seismological Press, Beijing (in Chinese).

Zhang, Y., W. P. Feng, L. S. Xu, C. H. Zhou and Y. T. Chen (2009a). Spatio-temporal rupture process of the 2008 great Wenchuan earthquake, *Sci. China Earth Sci.*, 52 (2):145-154. https://doi.org/10.1007/s11430-008-0148-7.

Zhang. Z., Y. Wang, Y. Chen, et al. (2009b). Crustal structure across Longmenshan fault belt from passive source seismic profiling. *Geophys. Res. Lett.*, 36.